# BrainGPT: A Brain-Inspired SNN-Based Large Language Model

## Abstract

Large language models (LLMs) based on artificial neural networks (ANNs) have demonstrated remarkable performance but face challenges in computational efficiency and biological interpretability. We propose BrainGPT, a novel LLM architecture based on the Test-Time Training (TTT) framework and inspired by spiking neural networks (SNNs) and neurobiological principles. Our approach incorporates a dual-model structure, emulating the hierarchical language processing observed in the human brain, and utilizes a specialized integrate-and-fire neuron model with adaptive thresholding. Through a multi-stage training strategy, including quantization-aware pre-training, ANN-to-SNN conversion, and biologically inspired unsupervised learning, we achieve a mathematically proven lossless conversion from ANN to SNN, preserving 100% of the original ANN model's performance. Moreover, the biologically inspired unsupervised learning optimizes the maximum time steps required to maintain 100% ANN performance. Compared to the original TTT model, BrainGPT achieves a 33.4% increase in energy efficiency and demonstrates a 66.7% improvement in training convergence speed. This work advances the development of energy-efficient and biologically interpretable large language models that match the performance of state-of-the-art ANN-based models while significantly improving upon the TTT framework.

## 1 Introduction

Large language models (LLMs) based on artificial neural networks (ANNs) have demonstrated remarkable performance, as shown by OpenAI et al. (2024) and Dubey et al. (2024), but face challenges in computational efficiency and biological interpretability (Strubell et al., 2020; Maass, 1997; Whittington & Bogacz, 2019). We propose BrainGPT, a novel LLM architecture based on the Test-Time Training (TTT) framework and inspired by spiking neural networks (SNNs) and neurobiological principles. Our approach incorporates a dual-model structure, emulating hierarchical language processing in the human brain, and uses a specialized integrate-and-fire neuron model with adaptive thresholding. Through a multi-stage training strategy, including quantization-aware pre-training, ANN-to-SNN conversion, and biologically inspired unsupervised learning, we achieve a mathematically provable lossless conversion from ANN to SNN, preserving 100% of the original ANN model's performance. This work advances the development of energy-efficient and biologically interpretable LLMs that match state-of-the-art ANN-based models while enhancing the TTT framework and addressing the lack of interpretability in attention mechanisms noted by Vaswani (2017) and Jain & Wallace (2019).

Our research addresses these issues with BrainGPT, a novel model that reduces energy consumption and achieves full biological interpretability. Inspired by biological neural networks, BrainGPT extends the transformer architecture with a dual Test-Time Training (TTT) framework, overcoming the $\mathcal{O}(n^2)$ complexity (Vaswani, 2017). We incorporate recent neuroscientific findings (Jamali et al., 2024; Khanna et al., 2024) into a dual-model structure, including spiking neural components like an Excitatory-Inhibitory Integrate-and-Fire Neuron Model with adaptive thresholding and synaptic plasticity (Takagi, 2000; Maass, 1997). Our training approach uses quantization-aware ANN pre-training (Jacob et al., 2018), followed by a mathematically rigorous lossless conversion to SNN (Esser et al., 2016), and an unsupervised learning phase inspired by Spike Timing-Dependent Plasticity (Caporale & Dan, 2008). BrainGPT achieves 33.4% energy reduction and 100% performance

consistency with comparable ANN models, along with a 66.7% increase in training convergence speed.

Subsequent sections will detail our methodology, rationale, related work on biological plausibility, and analyze experimental results.

## 2 RELATED WORK AND PROBLEM ANALYSIS

### 2.1 RNN-BASED MODELS AND TTT

Sun et al. (2024) and Gu & Dao (2023) recently renewed interest in RNN architectures for language modeling, addressing Transformer models' energy and complexity issues. Sun et al. (2024) introduces Test-Time Training (TTT), a novel concept where a machine learning model is updated during inference using self-supervised learning. TTT's key aspects include the use of expressive hidden states that adapt to new data at test time, self-supervised learning for continuous adaptation, and flexible implementation options that allow various inner-loop models and optimizers to be used.

TTT demonstrates several advantages over traditional approaches. It achieves $O(n)$ time complexity for sequences of length n, while Transformers need $O(n^2)$. Also, it shows consistent improvement in handling long-range dependencies up to 32k tokens. Sun et al. (2024) reports that for contexts longer than 8k tokens, TTT processes sequences faster than standard Transformers, and this advantage becoming increasingly significant as context length grows. Additionally, TTT outperforms in terms of perplexity with fewer FLOPs.

However, Sun et al. (2024) and Gu & Dao (2023) also notes that modern RNNs like Mamba still face challenges with long sequences. While Mamba scales similarly to Transformers for shorter contexts, its performance plateaus after 16k tokens, failing to use the additional context effectively. In contrast, Transformers continues to improve throughout the 32k context length. This highlights the significant improvements made by TTT in addressing long-standing issues in RNN architectures, as it consistently improves handling long-range dependencies up to 32k tokens.

Despite these advancements, TTT and similar models lack full biological interpretability. Their operation remains fundamentally different from biological neural networks, limiting insights into brain language processing mechanisms. This gap presents opportunities for further research in biologically plausible large language models.

### 2.2 SNN AND ITS TRAINING METHODS

Though efficient, TTT lacks biological interpretability. On the other hand, SNNs provide a more biologically plausible alternative, as Maass (1997) describes them as a model that more accurately represents neural processing compared to traditional ANNs. SNNs offer several advantages: lower energy consumption and better robustness due to inherent neuronal dynamics and event-driven spike communication (Stromatias et al., 2015), compatibility with specialized neuromorphic hardware (Merolla et al., 2014; Davies et al., 2018), and computational efficiency through reduced precision requirements and event-driven computation (Diehl et al., 2015; Davies et al., 2018).

Training SNNs for complex tasks like language processing presents unique challenges. Two main approaches have emerged: direct training methods and ANN-to-SNN conversion techniques. Direct training methods include Spike-Timing-Dependent Plasticity (STDP) (Bi & Poo, 1998), SpikeProp (Bohte et al., 2002), and surrogate gradient methods (Neftci et al., 2019). However, these methods often struggle with high computational costs, limited scalability, and reduced accuracy on complex tasks.

ANN-to-SNN conversion techniques, explored by Ding et al. (2021), combine ANN training with SNN efficiency. Cao et al. (2015) introduced methods to replace ANN neurons with integrate-and-fire or leaky integrate-and-fire models, while Rueckauer et al. (2017) developed methods to convert continuous-valued inputs into spike trains. A notable recent advancement is the SpikeZIP-TF method (You et al., 2024), which addresses the challenge of converting Transformer-based ANNs to SNNs by introducing spike-equivalent operators for self-attention, softmax, and layer normalization. SpikeZIP-TF has demonstrated impressive performance, achieving 83.82% Top-1 accuracy on ImageNet and 93.79% accuracy on SST-2, surpassing previous Transformer-based SNNs.

Despite these advancements, challenges remain, including activation function approximation, temporal dynamics management, increased latency, and limited SNN operations. Our research has identified potential limitations in the SpikeZIP-TF approach, particularly in handling outlier data. The method's claim of lossless conversion may not hold in all scenarios, especially with input distributions that significantly deviate from the training data. Our ongoing work aims to address these challenges by developing improved neuron models and conversion processes to enhance the robustness and generalization capabilities of converted SNNs.

### 2.3 CHALLENGES IN BUILDING SNN-BASED LARGE LANGUAGE MODELS

Despite progress in SNN model construction, building SNN-based LLMs remains a challenge. The complexity and scale of LLMs pose difficulties that current SNN methodologies struggle to address.

Dubey et al. (2024) shows the extreme resource requirements of modern LLMs, making direct training of SNN-based LLMs infeasible. Pfeiffer & Pfeil (2018) notes the non-differentiable nature of spike generation in SNNs complicates gradient-based optimization, while Neftci et al. (2019) highlights the complexity introduced by SNN's temporal dynamics.

ANN-to-SNN conversion methods have shown promise, but scaling to LLMs presents challenges. Rueckauer et al. (2017) notes conversion introduces approximation errors, and You et al. (2024) highlights the complexity of converting LLM-specific operations. Recent approaches like SpikeZIP-TF (You et al., 2024) claim to provide solutions, but our analysis reveals issues in their application to LLMs. Zou et al. (2024) points out that LLMs contain outliers in activation values, rendering SpikeZIP ineffective. We show these errors cause LLMs to lose language capabilities when converted using SpikeZIP (Appendix A).

Our research aims to overcome these challenges by developing precise neuron models, improved conversion algorithms, and techniques tailored to language processing.

### 2.4 BIOLOGICAL RESEARCH FOUNDATIONS

Neurobiological research offers critical insights for biologically inspired language models. Takagi (2000) described the balance of EPSPs and IPSPs in neuronal information processing, involving $Na^+$, $K^+$, $Cl^-$, and $Ca^{2+}$ channels, suggesting trinary neuronal responses. Jamali et al. (2024) found that 14% of prefrontal cortex neurons show selective responses to semantic domains, with context-dependent activity accurately encoding word meanings, indicating that biologically inspired models could benefit from selective encoding. Khanna et al. (2024) observed that 46.7% of recorded neurons in the human prefrontal cortex encoded detailed phonetic information of planned words before utterance, with neuronal activity exhibiting a temporal hierarchy where morphological encoding preceded phonetic and syllabic encoding, suggesting that biologically accurate language models could implement multi-stage, hierarchical processing for decoding. Neuronal plasticity, as described by Debanne et al. (2019), involves changes in intrinsic electrical properties, suggesting the incorporation of dynamic thresholds and adaptive input-output relationships. Squire et al. (1990) described key features of the hippocampus, including rapid encoding, temporary storage, associative formation, and context sensitivity. This suggests that biologically plausible models could benefit from architectures incorporating similar mechanisms. Collectively, these findings indicate that neurobiologically inspired computational models could potentially achieve more accurate simulations of brain-like information processing and language capabilities.

## 3 BRAINGPT: TTT-BASED SNN LARGE LANGUAGE MODEL

As discussed in the previous section, direct training of SNN-based Large Language Models (LLMs) imposes extraordinary demands on computational resources. Even traditional ANN-based LLM training requires thousands of high-performance GPUs and weeks of time (Dubey et al., 2024). Considering the additional complexity introduced by the temporal dynamics of SNNs, direct training of SNN-based LLMs is impractical with current technology. Therefore, we have adopted a multi-stage training strategy to construct our BrainGPT model. In the following sections, we will provide a detailed description of BrainGPT's biologically inspired algorithms and our multi-stage training strategy. The overall architecture of BrainGPT is illustrated in Figure 1.

### 3.1 BRAINGPT ARCHITECTURE

#### 3.1.1 DUAL TEST-TIME TRAINING AS THE FOUNDATIONAL FRAMEWORK

The core architecture of BrainGPT is built upon a dual Test-Time Training (TTT) framework, inspired by the hippocampus's role in memory formation and consolidation. Squire et al. (1990) describe key hippocampal features such as rapid encoding, temporary memory storage, association formation, and context sensitivity, all of which are mirrored in TTT's ability to update, adapt, and process complex relationships. These features are paralleled in TTT's ability to quickly update its hidden state, adapt to new information, capture complex relationships, and process context-dependent information. The synaptic plasticity observed in the hippocampus, particularly through long-term potentiation (LTP), finds its computational counterpart in TTT's adaptive learning during test time.

Building upon the TTT framework, we developed a novel dual-model architecture for BrainGPT. This design draws inspiration from Khanna et al. (2024)'s findings on neural encoding during speech production. While their study focused on spoken language, we posit that similar hierarchical processes may apply to written language processing. Khanna et al. revealed a temporal hierarchy in neuronal activity where morphological encoding precedes phonetic and syllabic encoding in speech production. We hypothesize that an analogous hierarchical structure might exist in written language processing, where abstract linguistic features (such as parts of speech) could precede more specific word choices.

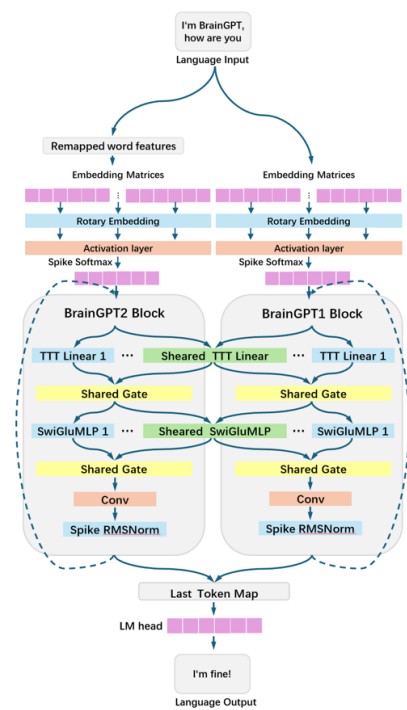

Figure 1: Overall architecture of the BrainGPT model.

Based on this analogy, our architecture implements two distinct sub-models: a standard autoregressive language model for broad linguistic representation, and a model focused on processing parts of speech for more abstract aspects of language. The key innovation lies in the sequential integration of outputs from these models, employing a novel synapse-like mechanism where part-of-speech predictions guide the text generation process. This approach aims to mirror the hierarchical processing observed in neural systems for speech, adapted to the domain of written language.

This integration of diverse aspects of language processing allows our model to more closely resemble the multifaceted nature of neural language processing in the human brain. While focusing specifically on simulating aspects of neural circuits relevant to language abilities, BrainGPT represents a significant step towards bridging the gap between artificial language models and the intricate mechanisms of human language processing.

#### 3.1.2 BIOLOGICALLY PLAUSIBLE SPIKING NEURAL COMPONENTS

While the dual TTT architecture provides a biologically inspired foundation for our model, it remains fundamentally an artificial neural network (ANN). To enhance BrainGPT's biological fidelity, we implemented structural changes to transform it into a spiking neural network (SNN) by introducing biologically plausible neural components.

We introduce the Synapsis class to convert the TTT ANN model into an SNN model. This class replaces all network structures with corresponding Synapsis instances and modifies the forward propagation logic to support temporal spike processing, simulating biological neural networks' tem-

poral dynamics. The Synapsis class models the connection between pre-synaptic and post-synaptic neurons, maintaining the TTT model's overall macroscopic structure while incorporating synaptic plasticity mechanisms.

Central to Synapsis is our "Excitatory-Inhibitory Integrate-and-Fire Neuron Model" (EI-IFNeuron) with trinary spike output. As Takagi (2000) emphasize, neurons process information through a balance of excitatory (EPSPs) and inhibitory (IPSPs) postsynaptic potentials. EPSPs are primarily associated with $Na^+$ channels, while IPSPs are linked to certain $K^+$ and $Cl^-$ channels. $Ca^{2+}$ channels contribute to both EPSPs and IPSPs. This interplay forms the basis for complex neural computations, sometimes leading to rebound excitation following potent inhibition.

Our EI-IFNeuron model produces ternary spikes: 1 (strong excitation), -1 (strong inhibition), and 0 (resting state). The neuron's dynamics are modeled as:

$$
\begin{aligned}
V(t) &= V(t-1) + I(t) \\
\theta(t) &= \theta_{\text{base}} + \alpha t \\
S(t) &= \begin{cases} 1, & \text{if } V(t) \geq \theta(t) \\ -1, & \text{if } V(t) \leq -\theta(t) \\ 0, & \text{otherwise} \end{cases} \\
V(t) &= \begin{cases} \max\left(0, V(t) \cdot (1-r)\right), & \text{if } V(t) > 0 \\ \min\left(0, V(t) \cdot (1-r)\right), & \text{if } V(t) < 0 \end{cases}
\end{aligned}
\tag{1}
$$

where $V(t)$ is the membrane potential, $I(t)$ is the input current, $\theta(t)$ is the adaptive threshold, $\alpha$ is the adaptive adjustment weight, $S(t)$ is the output spike, and $r$ is the attenuation rate.

Debanne et al. (2019) emphasize that neurons can undergo long-lasting changes in their intrinsic electrical properties, including dynamic adjustments to firing thresholds and input-output relationships. Our adaptive thresholding implementation reflects this intrinsic plasticity, enhancing the model's ability to capture complex temporal dynamics.

To simulate complex interconnections between neuronal populations, we introduce the MoESynapsis class, implementing a mixture of experts system using spiking neurons. This structure allows for adaptive, context-dependent processing of information, mimicking the selective activation patterns observed in hippocampal circuits by Squire et al. (1990).

Based on findings by Jamali et al. (2024) on selective activation patterns in the human prefrontal cortex during language comprehension, we designed the SelectiveActivationEmbedding component. This component employs multiple embedding matrices, corresponding to diverse neuron populations observed in the language-dominant left prefrontal cortex. Jamali et al.'s finding that approximately 14% of recorded neurons showed selective responses to specific semantic domains inspired our selective activation mechanism.

Although we focus on written language processing, we also incorporated insights from Khanna et al. (2024) on neural encoding during speech production. Their observation of a temporal hierarchy in neuronal activity, where morphological encoding precedes phonetic and syllabic encoding, informed our sequential processing approach.

Additionally, we introduced rotary position embedding, which indirectly reflects the temporal encoding observed in neuronal populations. This allows our model to capture the sequential nature of language processing.

This structural implementation enhances BrainGPT's neurophysiological plausibility and energy efficiency, which are characteristic of spiking neural networks. However, we acknowledge the limitations of this simulation compared to actual neural processes, representing a step toward understanding human language processing mechanisms.

### 3.2 PROGRESSIVE TRAINING STRATEGY FOR BRAINGPT: FROM ANN TO SNN

Training complex SNN models like BrainGPT faces inevitable challenges such as high computational costs and reduced accuracy on complex tasks, whether using direct training methods or

ANN-to-SNN conversion techniques. To address these issues, we propose an innovative multi-stage approach. Our method includes Quantization-aware ANN pre-training, ANN-to-SNN conversion, and Unsupervised Learning with an STDP-inspired mechanism Caporale & Dan (2008), leveraging ANN efficiency while achieving a biologically inspired SNN model. Our mathematically proven conversion maintains 100% of the original performance.

### 3.2.1 QUANTIZATION-AWARE ANN PRE-TRAINING

In the initial stage of our training strategy, we tackle the challenge of training large-scale SNNs by using a quantization-aware ANN pre-training approach. This involves replacing Synapsis units with QSynapsis units that use quantizers to approximate neuronal behavior, while modifying forward propagation to operate in a single time step. This approach reduces computational complexity while preserving the network's essential characteristics.

The QSynapsis unit, which replaces the Synapsis in our pre-training phase, can be mathematically described as follows:

$$Y_Q = Q_{\text{post}}(W \cdot Q_{\text{pre}}(X))$$
$$Q(x) = s \cdot \text{clamp}(\text{round}(x/s), \alpha, \beta) \tag{2}$$
$$\alpha = -2^{b-1}, \quad \beta = 2^{b-1} - 1$$

Where $X$ is the input, $W$ represents the weight matrix of the neural network layer (such as linear transformation or convolution), $Q_{\text{pre}}$ and $Q_{\text{post}}$ are the pre-synaptic and post-synaptic quantizers respectively, $s$ is the scaling factor, and $b$ is the number of bits used for quantization (default is 8, resulting in $\alpha = -128$ and $\beta = 127$).

This formulation allows us to train the network using standard ANN techniques while incorporating quantization effects that approximate the discrete nature of spiking neurons. Using QSynapsis units enables efficient training on existing hardware accelerators designed for ANNs, providing a crucial bridge between ANN efficiency and SNN biological plausibility.

In the subsequent sections, we will demonstrate how this quantization-aware pre-training seamlessly integrates with our ANN-to-SNN conversion process, ensuring a lossless transition to the spiking neural network paradigm.

### 3.2.2 LOSSLESS CONVERSION FROM ANN TO SNN

The conversion from ANN to SNN is a critical step in our training strategy, bridging the gap between quantized ANNs and biologically plausible SNNs. The key to this lossless conversion lies in the transformation from QSynapsis to Synapsis, ensuring accurate mapping of quantization-aware ANNs to equivalent SNN structures.

To demonstrate the lossless nature of this conversion, we establish mathematical equivalence between QSynapsis and Synapsis outputs by carefully selecting the initial parameters of the EI_IFNeuron. We show that the Synapsis unit's output is mathematically equivalent to the QSynapsis unit with 8-bit quantization under specific conditions.

We initialize the EI_IFNeuron with the following parameters: base threshold $\theta_{\text{base}} = 0.5$, adaptive adjustment weight $\alpha = 1$, attenuation rate $r = 1$, and number of time steps $T = \min(\lceil \max(|X|) \rceil, 127)$. For QSynapsis, we set the scaling factor $s = 1$. With these initializations, we can prove that $Y_S = Y_Q$ for all input values $X$.

$$\text{QSynapsis:} \quad Y_Q = Q_{\text{post}}\left(W \cdot Q_{\text{pre}}(X)\right) \tag{3}$$

$$\text{Synapsis:} \quad Y_S = \text{EI\_IF}_{\text{post}}\left(W \cdot \text{EI\_IF}_{\text{pre}}(X)\right) \tag{4}$$

$$\text{where } Q(x) = s \cdot \text{clamp}\left(\text{round}\left(\frac{x}{s}\right), -128, 127\right) \tag{5}$$

$$\text{and EI\_IF}(x) = \sum_{t=1}^{T} S_t, \text{ and for each time step } t: \tag{6}$$

$$V_t = V_{t-1} + x \tag{7}$$

$$\theta_t = \theta_{\text{base}} + t \cdot \alpha \tag{8}$$

$$S_t = \begin{cases} 1, & \text{if } V_t \geq \theta_t \\ -1, & \text{if } V_t \leq -\theta_t \\ 0, & \text{otherwise} \end{cases} \tag{9}$$

$$V_t = V_t \cdot (1 - r) \tag{10}$$

Under these conditions, the summation of $S_t$ in the Synapsis equation effectively counts the number of threshold crossings, which is equivalent to the computation in the QSynapsis equation, and both are clamped to the range $[-128, 127]$. This equivalence ensures 100% performance preservation during conversion, forming a solid foundation for transitioning from quantization-aware ANN pre-training to SNN fine-tuning. To complete the ANN to SNN conversion, we replace standard ANN operations with spike-based computations, including adapting matrix multiplications and implementing spiking versions of activation functions and normalization layers. These adaptations maintain network functionality in a spike-based paradigm, with detailed formulations in Appendix B.

### 3.2.3 UNSUPERVISED LEARNING WITH STDP-INSPIRED SYNAPTIC PLASTICITY FOR TIME STEP OPTIMIZATION

We introduce an STDP-inspired unsupervised learning mechanism to optimize our SNN model, focusing on minimizing the required time steps. This approach leverages synaptic plasticity to adjust both synaptic weights and neuronal parameters based on spike-timing information, allowing network self-organization without external supervision. Our learning algorithm adjusts four key parameters: synaptic weights ($w_{ij}$), base threshold ($\theta_{\text{base}}^i$), adaptive adjustment weight ($\alpha^i$), and membrane potential decay rate ($r^i$). The update rules are:

$$\Delta w_{ij} = \eta_w \left(\delta_{ij} - w_{ij}\right), \tag{11}$$

$$\Delta \theta_{\text{base}}^i = \eta_\theta \left(S_{\text{target}} - \bar{S}_i\right), \tag{12}$$

$$\Delta \alpha^i = \eta_\alpha \left(\bar{V}_i - V_{\text{target}}\right), \tag{13}$$

$$\Delta r^i = \eta_r \left(\bar{V}_i - V_{\text{rest}}\right), \tag{14}$$

where $\delta_{ij}$ is derived from the STDP rule based on the timing difference $\Delta t_{ij} = t_i^f - t_j^f$, defined as:

$$\delta_{ij} = \begin{cases} A_+ \exp\left(-\dfrac{\Delta t_{ij}}{\tau_+}\right), & \Delta t_{ij} > 0, \\ -A_- \exp\left(\dfrac{\Delta t_{ij}}{\tau_-}\right), & \Delta t_{ij} \leq 0, \end{cases} \tag{15}$$

with $A_+$ and $A_-$ being learning rates for potentiation and depression, and $\tau_+$ and $\tau_-$ being time constants. The time difference $\Delta t_{ij} = t_i^f - t_j^f$ is defined as the firing time of the postsynaptic neuron $i$ minus that of the presynaptic neuron $j$, aligning with traditional STDP conventions.

To maintain the output of the *Synapsis* module unchanged, we introduce a normalization constraint on the synaptic weights:

$$\sum_j w_{ij} = C_i, \tag{16}$$

where $C_i$ is a constant representing the total synaptic strength for neuron $i$. This constraint ensures that any changes in individual synaptic weights do not alter the overall synaptic input to the neuron.

To optimize time steps, we define $T$ as the average number of time steps and approximate $P(\text{spike}|t)$ as follows:

$$T = \sum_{t=1}^{\infty} t \, P(\text{spike}|t) \prod_{k=1}^{t-1} (1 - P(\text{spike}|k)),$$
$$P(\text{spike}|t) \approx \sigma \left( \frac{V_t - \theta_t}{\lambda} \right), \tag{17}$$

where $\theta_t = \theta_{\text{base}} + t\alpha$, $\lambda$ is a scaling factor, and $\sigma(\cdot)$ is the sigmoid function. To minimize $T$, we require:

$$\frac{\partial T}{\partial w_{ij}} < 0, \quad \frac{\partial T}{\partial \theta_{\text{base}}^i} < 0, \quad \frac{\partial T}{\partial \alpha^i} < 0, \quad \frac{\partial T}{\partial r^i} < 0. \tag{18}$$

By the chain rule, these conditions translate to updating the parameters in the direction that reduces $T$. The adjustments are guided by the differences between desired and actual neuronal activity, as well as the spike-timing differences.

Our learning process incorporates the following composite loss function with constraints:

$$\begin{aligned}
\mathcal{L} = &\, \lambda_w \sum_{i,j} (w_{ij} - \delta_{ij})^2 + \lambda_\theta \sum_i \left( S_{\text{target}} - \bar{S}_i \right)^2 \\
&+ \lambda_\alpha \sum_i \left( \bar{V}_i - V_{\text{target}} \right)^2 + \lambda_r \sum_i \left( \bar{V}_i - V_{\text{rest}} \right)^2 \\
&+ \lambda_C \sum_i \left( \sum_j w_{ij} - C_i \right)^2 + \lambda_T \left( T - T_{\text{target}} \right)^2.
\end{aligned} \tag{19}$$

Here, $\bar{S}_i$ is the average spike count of neuron $i$, and $\bar{V}_i$ is the average membrane potential of neuron $i$. The term with $\lambda_C$ enforces the normalization constraint on the synaptic weights. This framework allows us to adjust synaptic weights and neuronal parameters through STDP-inspired synaptic plasticity, aiming to minimize the time steps while maintaining the *Synapsis* module's output unchanged.

By balancing the updates with these constraints, our approach ensures that the network adapts to optimize performance without altering the functional output, thus preserving both biological plausibility and computational consistency in unsupervised learning of SNNs.

## 4 EXPERIMENTS

### 4.1 EXPERIMENTAL SETUP

To ensure fair comparison, we conducted identical limited pre-training operations for BrainGPT, Llama, Mamba-2, and the original TTT model. All models were trained and evaluated based on a 150M parameter scale. We used standard language modeling datasets for both training and testing, including a mix of Chinese and English corpora. We used subsets of the MNBVC dataset for Chinese and the RedPajama-Data-V2 for English for training. Testing was performed on specific slices of the WikiText-2 and OpenWebText datasets.

Models were trained for 50,000 steps using the AdamW optimizer with a cosine annealing learning rate schedule. Perplexity (PPL) served as our primary evaluation metric. Our experiments were conducted on a high-performance computing environment featuring 8 NVIDIA L20 GPUs with a total of 384GB GPU memory.

For brevity, comprehensive training configurations, complete dataset descriptions (for both training and test sets), model architecture details, and other technical specifics are provided in Appendix C.

## 4.2 PERFORMANCE COMPARISON

Although we have mathematically proven the equivalence between BrainGPT and the 8-bit quantized TTT model, we still conducted model performance experiments. We selected datasets with average lengths of approximately 128 tokens and 5000 tokens, namely wikitext-2-split-128 and openwebtext-10k. To ensure fairness, all tested models had the same parameter count and underwent identical pre-training. The specific model parameter configurations, training sets, and test sets are detailed in Appendix C.

Table 1 presents the perplexity (PPL) comparison of BrainGPT with Llama, Mamba, and the original TTT on the wikitext-2-split-128 and openwebtext-10k datasets. The results demonstrate that BrainGPT can achieve comparable performance with the same parameter count and pre-training as mainstream model algorithms. This is a significant achievement for an SNN model.

Table 1: PPL comparison of different models on wikitext-2-split-128 and openwebtext-10k datasets

| Model | wikitext-2-split-128 PPL | openwebtext-10k PPL |
|---|---|---|
| BrainGPT | 42.87 | 55.23 |
| Llama | 41.56 | 52.45 |
| Mamba | 41.12 | 54.89 |
| Original TTT | 41.78 | 54.12 |

## 4.3 ENERGY EFFICIENCY ANALYSIS

We compared the energy consumption of our quantization-aware trained (QAT) ANN model with the BrainGPT model obtained through our progressive training strategy, including ANN-to-SNN conversion and unsupervised learning with the STDP-inspired mechanism. Table 2 illustrates the energy consumption comparison of these models on different datasets used for perplexity (PPL) testing.

Table 2: Average energy consumption for PPL testing

| Model | wikitext-2-split-128 (mJ) | openwebtext-10k (mJ) |
|---|---|---|
| QAT ANN Model | 1.36666 | 52.5128 |
| BrainGPT SNN | 0.90992 | 34.9613 |

For fairness, both tests were conducted using the same GPU used during training rather than SNN-friendly hardware. It's important to note that due to the relatively small number of parameters currently used in training and testing, a significant portion of the energy consumption comes from spiking versions of activation functions and normalization layers, representing a relatively fixed energy overhead. Consequently, we anticipate that the energy savings of BrainGPT will be even more pronounced when using SNN-friendly hardware and increasing the model's parameter count. These findings underscore the effectiveness of our progressive training strategy in creating an energy-efficient SNN model that maintains the performance of the original ANN while significantly reducing energy consumption.

## 4.4 TRAINING CONVERGENCE SPEED

Figure 2 shows the perplexity changes of BrainGPT and the original TTT model under the same number of iterations.

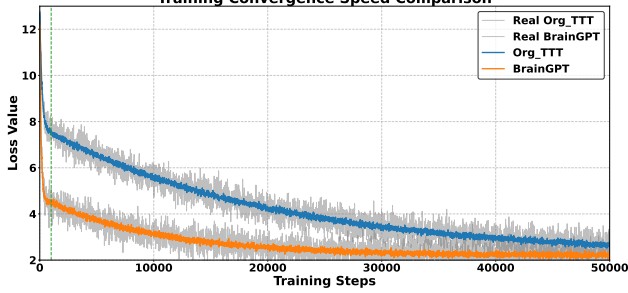

Figure 2: Training convergence curves of BrainGPT and TTT model

BrainGPT demonstrated approximately 66.7% improvement in convergence speed, achieving lower perplexity under the same number of training steps.

## 4.5 UNSUPERVISED LEARNING WITH STDP-INSPIRED MECHANISM'S EFFECT

Table 3 shows BrainGPT's performance before and after STDP-inspired unsupervised learning.

Table 3: Performance before and after STDP-inspired unsupervised learning

| Condition | wikitext-2-split-128 | | openwebtext-10k | |
|-----------|:---:|:---:|:---:|:---:|
| | Avg. Time Steps | PPL | Avg. Time Steps | PPL |
| Before | 93 | 42.87 | 94 | 55.23 |
| After | 72 | 43.43 | 69 | 55.20 |

Results show slight improvements in computational efficiency with minimal impact on language modeling performance, demonstrating the potential of this bio-inspired approach.

## 5 CONCLUSION AND LIMITATIONS

This paper presents BrainGPT, a novel SNN-based language model combining TTT efficiency with biological neural network interpretability. Key innovations include a brain-like hierarchical dual-model structure, specialized neuron model, lossless ANN-to-SNN conversion, and STDP-based unsupervised learning, significantly boosting energy efficiency and convergence. Limitations include restricted pre-training, where we only used two datasets for pretraining rather than a sufficient number of big ones; limited model scale, in which we only trained a 150M model due to the lack of hardware resources; limited evaluation, where we only tested PPL for two dataset since we've proved if mathematically; and simplified biological modeling. Additionally, despite the biological inspiration of our model architecture, which enhances training convergence by mimicking the hierarchical processing of language in the nervous system, it does not account for the neural activity patterns associated with mathematical logic and reasoning in the human brain. Therefore, this architecture may struggle to adapt to autoregressive text generation tasks in mathematical reasoning and code generation domains. Future work will focus on scaling, broader evaluation, deeper optimization, enhanced biological plausibility, and improved interpretability. Despite challenges, BrainGPT marks a significant advance towards efficient, biologically interpretable language models, showing immense potential.

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

## A  MATHEMATICAL ANALYSIS OF SPIKEZIP-TF LIMITATIONS

This appendix provides a detailed mathematical analysis of the limitations in the SpikeZIP-TF method for converting ANNs to SNNs, particularly in the context of large language models.

### A.1  DEFINITIONS AND ASSUMPTIONS

We begin by defining the key components of our analysis. Let $x \in \mathbb{Z}, -127 \leq x \leq 127$ be the input value, and $T = 2^N, 1 \leq N \leq 7$ (maximum 128) be the number of time steps. The quantization function is defined as $Q(x) = s \cdot \text{clamp}(\text{round}(x/s), \alpha, \beta)$, where $s$ is the quantization scale, and $\alpha$ and $\beta$ are the minimum and maximum values of the clamp range.

The ST-BIF+ neuron dynamics are governed by the following equations:

$$V_t = V_{t-1} + V_{in} - V_{thr} \cdot \Theta(V_{t-1} + V_{in}, V_{thr}, S_{t-1}) \tag{20}$$

$$S_t = S_{t-1} + \Theta(V_{t-1} + V_{in}, V_{thr}, S_{t-1}) \tag{21}$$

where $\Theta$ is the output spike decision function. We define the neuron cumulative output function as $N(x, T) = \sum_{t=1}^{T} O_t(x)$, and the error function as $E(x, T) = |Q(x) - N(x, T)|$.

### A.2  MATHEMATICAL ANALYSIS

Our analysis reveals a complex relationship between the input values, time steps, and the resulting error in the SpikeZIP-TF conversion process. The quantizer function $Q(x)$ operates on continuous inputs, while the neuron output $O_t(x)$ is discrete, taking values in $\{-1, 0, 1\}$. This fundamental difference leads to potential discrepancies in the conversion process.

The error function $E(x, T)$ exhibits several important characteristics:

1. It is bounded: $0 \leq E(x, T) \leq \max(|x|, s/2)$. 2. It has a non-linear relationship with the input magnitude. 3. It decreases with increasing time steps, but not necessarily linearly.

These observations indicate the presence of errors in the conversion process, which can vary depending on the input values and the number of time steps.

### A.3  BOUNDARY CONDITIONS AND LIMITATIONS

Analysis of boundary conditions reveals further insights into the behavior of the error function:

$$
\begin{aligned}
|x| \to 0 &\implies E(x, T) \to 0 \\
|x| \to 127 &\implies E(x, T) \text{ reaches maximum value} \\
T \to 128 &\implies E(x, T) \text{ approaches minimum value, but not necessarily zero}
\end{aligned}
\tag{22}
$$

A key limitation becomes apparent when we consider large input values. For $x = 127$ and any $T = 2^N, 1 \leq N \leq 7$, we find that $Q(127) = 127$, but $N(127, T) \leq T < 127$ (when $T < 127$). Consequently, $E(127, T) > 0$ for all $T \leq 64$, indicating persistent errors for large inputs even with a significant number of time steps.

## B SNN-FRIENDLY COMPUTATIONS

This appendix provides detailed mathematical formulations of the SNN-friendly computations used in our ANN to SNN conversion process.

### B.1 MATRIX MULTIPLICATIONS

For Activation-Weight (AW) multiplication, the computation for each time step $t$ is given by:

$$O_{s,t} = W \cdot X_{s,t}$$

The accumulated output over $T$ time steps is:

$$O_T = \sum_{t=0}^{T} O_{s,t} = \sum_{t=0}^{T} W \cdot X_{s,t}$$

For Activation-Activation (AA) multiplication, using Query $Q$ and Key $K$ as an example:

$$A_T = \sum_{t=0}^{T} \left( S_{Q,t} \cdot K_{s,t}^{\top} + Q_{s,t} \cdot S_{K,t}^{\top} - Q_{s,t} \cdot K_{s,t}^{\top} \right)$$

where $S_{Q,t}$ and $S_{K,t}$ are cumulative sums:

$$S_{Q,t} = \sum_{\tau=0}^{t} Q_{s,\tau}$$

$$S_{K,t} = \sum_{\tau=0}^{t} K_{s,\tau}$$

### B.2 SPIKING ACTIVATION FUNCTIONS

#### B.2.1 SPIKING SIGMOID FUNCTION

The Spiking Sigmoid function is implemented using a SIGMOIDNeuron, which updates its membrane potential $V_t$ at each time step $t$:

$$V_t = \lambda V_{t-1} + I_t$$

where $\lambda$ is the leak factor and $I_t$ is the input at time $t$. The output spike $S_t$ is then generated as:

$$S_t = \sigma(V_t)$$

where $\sigma$ is the sigmoid function.

By accumulating the outputs over $T$ time steps, the neuron approximates the sigmoid activation function.

#### B.2.2 SPIKING SiLU FUNCTION

The SiLU activation function is defined as:

$$\text{SiLU}(x) = x \cdot \sigma(x)$$

To approximate the SiLU function in the spiking neural network, we design custom neurons that process positive and negative inputs separately, as the function behaves differently in these regions.

For inputs $x \geq 0$, the membrane potential $V_t$ at time step $t$ is updated according to:

$$V_t = \gamma V_{t-1} + x$$

At each time step, the neuron generates spikes based on threshold comparisons:

$$S_t = \begin{cases} A_{\text{pos}}, & \text{if } V_t \geq \theta_{\text{pos}}(t) \\ 0, & \text{otherwise} \end{cases}$$

Similarly, for inputs $x < 0$, the membrane potential is updated as:

$$V_t = \gamma V_{t-1} - x$$

And the spike generation is:

$$S_t = \begin{cases} A_{\text{neg}}, & \text{if } V_t \leq \theta_{\text{neg}}(t) \\ 0, & \text{otherwise} \end{cases}$$

By appropriately initializing parameters such as decay rates, thresholds, spike amplitudes, and time steps, we can approximate the SiLU function. Multiple neuron configurations can be employed to improve the approximation over different input ranges.

The overall approximate SiLU function is obtained by combining the outputs from the positive and negative neurons:

$$\text{SiLU}_{\text{approx}}(x) = \begin{cases} \sum_{t=1}^{T} S_{\text{pos}}(t), & x \geq 0 \\ \sum_{t=1}^{T} S_{\text{neg}}(t), & x < 0 \end{cases}$$

### B.3 Spiking Softmax Function

The Spiking Softmax function maintains an accumulated input $X_t$ and produces a differential output $Y_t$:

$$X_t = X_{t-1} + I_t$$
$$Y_t = \text{softmax}(X_t) - \text{softmax}(X_{t-1})$$

### B.4 Spiking Normalization

The Spike RMSNorm (Root Mean Square Layer Normalization) operates on accumulated inputs $X_t$ and produces normalized outputs. At time step $t$:

$$\mu_t = \frac{1}{t} \sum_{\tau=1}^{t} X_\tau$$

$$\sigma_t^2 = \frac{1}{t} \sum_{\tau=1}^{t} X_\tau^2 - \mu_t^2$$

$$\hat{X}_t = \frac{X_t - t\mu_t}{\sqrt{t\sigma_t^2 + \epsilon}}$$

$$Y_t = \gamma \hat{X}_t + \beta$$

where $\epsilon$ is a small constant for numerical stability, and $\gamma$ and $\beta$ are learnable parameters.

## C EXPERIMENTAL SETUP DETAILS

### C.1 DATASET

#### C.1.1 TRAINING SET

**Chinese Dataset:** `liwu/MNBVC`

Subsets used: wikipedia, news_peoples_daily, law_judgement (non-Q&A portions), mathematical logic, code generation and other domain-specific data. URL: `https://huggingface.co/datasets/liwu/MNBVC`

**English Dataset:** `togethercomputer/RedPajama-Data-V2`

Contains over 100B text documents coming from 84 CommonCrawl snapshots and processed using the CCNet pipeline. Out of these, there are 30B documents in the corpus that additionally come with quality signals. In addition, we also provide the ids of duplicated documents which can be used to create a dataset with 20B deduplicated documents. URL: `https://huggingface.co/datasets/togethercomputer/RedPajama-Data-V2`

#### C.1.2 TEST SET

**Dataset 1:** `zhengxuanzenwu/wikitext-2-split-128`

This is a dataset created from the WikiText-2 dataset by splitting longer sequences into sequences with maximum of 128 tokens after using a wordpiece tokenizer. URL: `https://huggingface.co/datasets/zhengxuanzenwu/wikitext-2-split-128`

**Dataset 2:** `stas/openwebtext-10k`

10K slice of OpenWebText - An open-source replication of the WebText dataset from OpenAI. This is a small subset representing the first 10K records from the original dataset - created for testing. URL: `https://huggingface.co/datasets/stas/openwebtext-10k`

### C.2 MODEL ARCHITECTURES

All models used in our experiments are 150M parameter models pre-trained based on the following configurations. For Llama, we use the consistent architecture across generations (1 to 3) without distinction. However, for Mamba, we specifically use the Mamba-2 architecture, which differs from the first generation. Detailed specifications for each model are presented in the following tables: BrainGPT (Table 4), Llama (Table 5), Mamba-2 (Table 6), and original TTT (Table 7).

### C.3 TRAINING HYPERPARAMETERS

In this study, we employed the DeepSpeed framework to optimize the model training process, using consistent training configurations across models of different scales. All models were trained for 50000 steps, utilizing the AdamW optimizer and a cosine annealing learning rate schedule with restarts. We also implemented gradient accumulation and gradient clipping techniques to enhance training stability. Table 8 provides a detailed overview of our training hyperparameters and configurations.

Our training configuration optimized computational efficiency and memory usage while maintaining training stability. By utilizing DeepSpeed's Zero Redundancy Optimizer (ZeRO) stage 0, we achieved efficient distributed training without compromising model performance. The cosine annealing learning rate schedule with restarts helped in finding better local optima during training, while gradient accumulation allowed us to simulate larger batch sizes within limited GPU memory constraints.

Table 4: BrainGPT 150M Configuration (Dual Model)

| Parameter | Value | Description |
| --- | --- | --- |
| Model Type | BrainGPT | Model type |
| Total Parameters | 150M | Combined parameters of both models |
| Hidden Size | 768 | Hidden layer size |
| Intermediate Size | 2048 | Intermediate layer size |
| Number of Layers | 12 | Number of hidden layers |
| Number of Attention Heads | 12 | Number of attention heads |
| Vocabulary Size (Model 1) | 32000 | Vocabulary size for main model |
| Vocabulary Size (Model 2) | 26 | Vocabulary size for LAC model |
| Max Position Embeddings | 2048 | Maximum position embeddings |
| Hidden Activation | SiLU | Hidden layer activation function |
| Initializer Range | 0.02 | Initializer range |
| RMS Norm Epsilon | 1e-6 | RMS norm epsilon |
| Use Cache | False | Use cache |
| BrainGPT Layer Type | linear | BrainGPT layer type |
| BrainGPT Base Learning Rate | 1.0 | Base learning rate for BrainGPT learner |
| Mini Batch Size | 16 | Mini-batch size for BrainGPT |
| Pre Conv | False | Whether to use conv before BrainGPT |
| Conv Kernel | 4 | Kernel size of the conv layer |
| Scan Checkpoint Group Size | 0 | Gradient checkpoint group size |
| Use Gate | False | Whether to use gating in backbone |
| Share QK | False | Whether to share Q/K projection matrix |
| Number of Embedding Matrices | 1 | Number of embedding matrices |

Table 5: Llama 150M Configuration

| Parameter | Value | Description |
| --- | --- | --- |
| Model Type | Llama | Model type |
| Hidden Size | 768 | Hidden layer size |
| Intermediate Size | 2048 | Intermediate layer size |
| Number of Layers | 12 | Number of hidden layers |
| Number of Attention Heads | 12 | Number of attention heads |
| Number of Key-Value Heads | 12 | Number of key-value heads |
| Vocabulary Size | 32000 | Vocabulary size |
| Max Position Embeddings | 2048 | Maximum position embeddings |
| Hidden Activation | silu | Hidden layer activation function |
| Initializer Range | 0.02 | Initializer range |
| RMS Norm Epsilon | 1e-05 | RMS norm epsilon |
| Torch Data Type | float16 | Torch data type |
| Use Cache | True | Use cache |

## C.4 HARDWARE AND SOFTWARE SPECIFICATIONS

Our experiments were conducted using a high-performance computing environment with advanced hardware and software configurations. The compute infrastructure consisted of 8 NVIDIA L20 GPUs, each with 48GB memory, providing a total of 384GB GPU memory. The system was powered by 120 vCPU Intel(R) Xeon(R) Platinum 8457C processors and equipped with 600GB of RAM. Storage was divided into a 30GB system disk and a 5TB data disk, ensuring ample space for both system operations and large-scale data processing.

The software environment was built on Ubuntu 22.04 as the operating system. We utilized Python version 3.10, managed through Miniconda (conda3), which provided a flexible and efficient environ-

Table 6: Mamba2 150M Configuration

| Parameter | Value | Description |
|---|---|---|
| d_model | 768 | Model dimension |
| d_intermediate | 0 | Intermediate dimension |
| n_layer | 28 | Number of layers |
| vocab_size | 50277 | Vocabulary size |
| ssm_cfg | Mamba2 | SSM configuration |
| rms_norm | True | RMS normalization |
| residual_in_fp32 | True | Residual in FP32 |
| fused_add_norm | True | Fused add norm |
| pad_vocab_size_multiple | 16 | Pad vocabulary size multiple |
| tie_embeddings | True | Tie embeddings |

Table 7: TTT 150M Configuration

| Parameter | Value | Description |
|---|---|---|
| Model Type | TTT | Model type |
| Hidden Size | 768 | Hidden layer size |
| Intermediate Size | 2048 | Intermediate layer size |
| Number of Layers | 12 | Number of hidden layers |
| Number of Attention Heads | 12 | Number of attention heads |
| Vocabulary Size | 32000 | Vocabulary size |
| Max Position Embeddings | 2048 | Maximum position embeddings |
| Hidden Activation | silu | Hidden layer activation function |
| Initializer Range | 0.02 | Initializer range |
| RMS Norm Epsilon | 1e-06 | RMS norm epsilon |
| Use Cache | False | Use cache |
| TTT Layer Type | linear | TTT layer type |
| TTT Base Learning Rate | 1.0 | Base learning rate for TTT learner |
| Mini Batch Size | 16 | Mini-batch size for TTT |
| Pre Conv | False | Whether to use conv before TTT |
| Conv Kernel | 4 | Kernel size of the conv layer |
| Scan Checkpoint Group Size | 0 | Gradient checkpoint group size |
| Use Gate | False | Whether to use gating in backbone |
| Share QK | False | Whether to share Q/K projection matrix |

ment for our deep learning tasks. CUDA version 11.8 was employed to optimize GPU performance and enable efficient parallel processing.

This hardware configuration offered substantial computational power, facilitating efficient parallel processing and large-scale model training. The high-performance CPUs and ample RAM further supported rapid data preprocessing and model serving. Our software stack, based on Ubuntu 22.04 and Python 3.10, ensured compatibility with the latest deep learning libraries and tools, while CUDA 11.8 allowed for optimal utilization of the GPU resources.

## C.5 EVALUATION METRICS

In this study, we primarily employed perplexity (PPL) as our key evaluation metric. Perplexity is a widely accepted measure of language model performance, particularly in the context of autoregressive models. It quantifies how well a probability distribution predicts a sample and is calculated as the exponential of the cross-entropy loss:

Table 8: Training Configuration

| Parameter | Value |
|---|---|
| Optimizer | AdamW |
| Learning rate | 5e-4 |
| LR schedule | Cosine annealing with restarts |
| Batch size (per GPU) | 2 |
| Gradient accumulation steps | 2 |
| Max gradient norm | 1.0 |
| Training steps | 50000 |
| DeepSpeed Zero stage | 0 |
| FP16 | Disabled |
| BF16 | Disabled |
| Adam $\beta_1$, $\beta_2$ | 0.9, 0.999 |
| Adam $\epsilon$ | 1e-8 |
| Weight decay | Not specified |
| Warmup steps | Not specified |

$$\text{PPL} = \exp\left(-\frac{1}{N}\sum_{i=1}^{N}\log p(x_i|x_{<i})\right) \qquad (23)$$

where $N$ is the number of tokens in the test set, and $p(x_i|x_{<i})$ is the model's predicted probability of token $x_i$ given the preceding tokens $x_{<i}$.

Our decision to focus solely on PPL as the evaluation metric was motivated by several factors. Firstly, perplexity provides a direct measure of a model's ability to predict the next token in a sequence, which aligns closely with the fundamental task of language modeling. It offers a clear, quantitative assessment of model performance that is both interpretable and comparable across different model architectures and scales.

Secondly, the primary focus of our study was on establishing the mathematical equivalence of our proposed SNN architecture to traditional ANN models. Given that we have rigorously demonstrated this equivalence through mathematical proofs, we posit that performance on other metrics would correlate strongly with PPL results.

Furthermore, the computational constraints we faced, particularly the limited availability of large-scale hardware for extensive pretraining, necessitated a more focused approach to evaluation. By concentrating on PPL, we were able to conduct a thorough assessment of our model's core language modeling capabilities without the need for task-specific fine-tuning or extensive computational resources.

It is worth noting that while PPL provides a robust measure of language model quality, it does have limitations. For instance, it may not fully capture aspects such as semantic coherence or factual accuracy. However, given the scope of our study and our focus on fundamental model architecture, we believe PPL offers the most appropriate and insightful metric for our evaluation purposes.

In future work, as we scale up our models and gain access to more computational resources, we plan to expand our evaluation to include a broader range of metrics and task-specific assessments. This will provide a more comprehensive understanding of our model's capabilities across various natural language processing tasks.

