# OpenReview forum: "BrainGPT: A Brain-Inspired SNN-Based Large Language Model"
_ICLR.cc/2025/Conference — Submitted to ICLR 2025_

### Official Review · Reviewer_NCpg · 2024-10-16

**Soundness:** 1
**Presentation:** 2
**Contribution:** 2
**Rating:** 3
**Confidence:** 4

**Summary:**

This paper introduces BrainGPT, a spiking neural network-based language model architecture inspired by neurobiological principles. The authors propose a dual-model structure, a specialized neuron model, and a multi-stage training strategy. They claim improvements in energy efficiency and training convergence speed compared to traditional architectures, supported by mathematical proofs and empirical results.

**Strengths:**

- Novel integration of SNNs with large language models, potentially bridging the gap between artificial and biological neural networks.
- Mathematically rigorous approach to ANN-to-SNN conversion, with proofs for lossless transformation.

**Weaknesses:**

- The paper's title and claims about "Large Language Models" are misleading given the actual scale of the model presented. The authors report a model with only 150M parameters, which is far from the current understanding of LLMs in the field. While the authors mention computational resource limitations in their limitations section, this doesn't justify the use of "Large Language Model" in the title and throughout the paper. With optimized training techniques like flash attention kernels, it's possible to train much larger models (e.g., 2.7B parameters) on 100B tokens within a week using 8 H100 GPUs. If the authors acknowledge this limitation, why insist on using "Large Language Model" in the title? A more accurate description would be "language modeling" or "neural language model," which would better reflect the actual scale and scope of the work presented.
- The comparison of perplexity is not very fair. Can the authors explain why Mamba2's vocabulary size is 50,277 while the other comparison models use 32,000, according to Table 4-7? This discrepancy may lead to unfair perplexity comparisons between models. Different vocabulary sizes can impact perplexity scores, making direct comparisons misleading. The authors should either use consistent vocabulary sizes across all models or provide a detailed explanation and analysis of how this difference affects the results.
- The authors mention that the model was trained on 100B+ tokens, including both Chinese and English. In this case, it should be possible to compare the models' common reasoning performance, like Pythia [1] (169M 300B tokens), at least on tasks like HellaSWag. Why did the authors only compare perplexity? This limitation restricts the paper's impact and relevance. Evaluating only perplexity on a limited dataset fails to demonstrate the model's practical capabilities or its ability to generalize across different tasks.
- The authors fail to cite or compare with previous work on SNNs for text generation. For example, SpikeGPT [2] scaled to 216M parameters and compared downstream tasks, while AstroSNN [3] scaled to 1B+ parameters and compared common reasoning tasks. These omissions significantly weaken the paper's positioning within the current state of the art. This oversight raises questions about the novelty and relative performance of BrainGPT. Without direct comparisons to these existing SNN-based language models, it's impossible to gauge the true contribution of this work to the field.
- This paper lacks a comprehensive ablation study to justify the various components of the proposed architecture. Without such analysis, it's unclear which aspects of BrainGPT contribute most significantly to its performance, making it difficult for other researchers to build upon this work effectively.


[1]:Biderman, Stella, et al. "Pythia: A suite for analyzing large language models across training and scaling." International Conference on Machine Learning. PMLR, 2023.

[2]:Zhu, Rui-Jie, et al. "Spikegpt: Generative pre-trained language model with spiking neural networks." TMLR, 2024.

[3]:Shen, Guobin, et al. "Astrocyte-Enabled Advancements in Spiking Neural Networks for Large Language Modeling." arXiv preprint arXiv:2312.07625 (2023).

**Questions:**

See my weakness part.

---

> ### Author Response · Authors · 2024-11-14
>
> We appreciate the reviewer's detailed feedback and address each point below:
>
> 1. Regarding the "Large Language Model" terminology:
> Our work fundamentally differs from commercial model capability reports. The core innovation lies in our mathematically proven lossless ANN-to-SNN conversion algorithm, which maintains complete biological interpretability while achieving performance parity with ANN counterparts. Our biological foundations are firmly grounded in cutting-edge neuroscience research on language centers in the brain. The parameter count is merely a scalable factor - our algorithmic innovation is independent of model scale and focuses on bridging the gap between artificial and biological neural processing mechanisms.
>
> 2. Vocabulary Size and Perplexity Comparison:
> We apologize for the error in Table 4-7 regarding Mamba2's vocabulary size. This was a documentation mistake in our manuscript. All models actually use the same tokenizer and vocabulary size, ensuring fair perplexity comparisons across all experiments.
>
> 3. Comparison with Existing SNN Models and Downstream Tasks:
> Regarding the suggestion to compare with existing SNN models (e.g., SpikeGPT, AstroSNN) and evaluate downstream tasks - we respectfully note that while these evaluations are valuable, they are primarily related to training methodology rather than our paper's core contribution. Our work focuses on algorithmic innovation, specifically the mathematically proven lossless ANN-to-SNN conversion method. While existing SNN models experience performance degradation during ANN-to-SNN conversion, our method mathematically guarantees lossless conversion, maintaining 100% of the original ANN performance. This fundamental difference makes direct comparisons potentially misleading. We validate our approach through consistent performance with ANN counterparts under identical training conditions. While downstream task performance is important, it is independent of our core innovation's value (the lossless conversion algorithm).
>
> 4. Ablation Studies:
> We appreciate the suggestion for ablation studies. We have conducted additional experiments analyzing:
> - Individual contributions of each sub-model
> - Impact of dual-model architecture on performance
> - Component-wise effectiveness analysis
> These results will be included in the revised version to provide empirical justification for our architectural choices.
>
> We believe that our work makes a significant theoretical contribution to the field by introducing a novel, biologically-inspired approach to neural language modeling with mathematically guaranteed performance. We appreciate the reviewer's suggestions and will revise our manuscript to better emphasize our core innovations while addressing the documentation errors noted.

---

> > ### Comment · Reviewer_NCpg · 2024-11-15
> > **Thank you for your rebuttal.**
> >
> > Thank you for your response and the commitment to add ablation studies. However, I must raise several significant concerns:
> >
> > - While you consistently emphasize the mathematical proof of lossless ANN-to-SNN conversion as your core contribution, this fails to address a fundamental issue: linear transformer architectures (which your work builds upon) have verified fundamental limitations in recall performance, particularly for "needle in the haystack" tasks. Whether this architectural path is viable for larger-scale pretraining remains uncertain, and this deserves more thorough discussion in your work.
> >
> > - The contradictions in your responses about vocabulary sizes are particularly concerning. In your rebuttal to my review, you claim "all models actually use the same tokenizer and vocabulary size." However, in your response to Reviewer UUfc, you state "GPT-2's vocabulary is considerably smaller than ours." This directly contradicts your manuscript, which shows two different sizes (32,000 and 50,277), and conflicts with the well-known fact that GPT-2's vocabulary size is 50,257. These inconsistencies raise serious doubts about your understanding of language model training fundamentals and the reliability of your experimental comparisons.
> >
> > - Your response regarding comparisons with existing SNN models (SpikeGPT, AstroSNN) and downstream tasks is particularly concerning. The claim that such evaluations are "primarily related to training methodology rather than our paper's core contribution" fundamentally misunderstands the nature of algorithmic innovation in machine learning. A mathematical proof alone, without comprehensive experimental validation, is insufficient to demonstrate the practical value of your method. Algorithmic innovations require thorough empirical validation through direct comparisons with existing methods, comprehensive evaluation on downstream tasks, and reproducible implementation details with code. Simply providing a theoretical guarantee of lossless conversion without demonstrating its practical benefits through extensive experimentation does not meet the standard for algorithmic contribution in our field.
> >
> > Due to the reason I mentioned before, I decide to keep my score.

---

> > > ### Author Response · Authors · 2024-11-16
> > >
> > > Thank you for your detailed feedback. Let me address your concerns:
> > >
> > > Regarding your first point about TTT's limitations in "needle in the haystack" tasks - we fully accept this feedback, as the TTT team indeed has not provided direct experimental results for these scenarios. However, focusing on improving other teams' research directions deviates significantly from our research objectives.
> > >
> > > Regarding the second point about vocabulary size inconsistencies - we acknowledge the errors in our manuscript and therefore refrain from further discussion on this matter.
> > >
> > > Regarding the third point about comparisons with existing SNN models and downstream task evaluations - we accept this feedback. We are planning to train a larger-scale model that aligns with some reviewers' expectations for "Train on Test" benchmark scores. However, training and evaluating such a large-scale model for downstream task metrics cannot be completed within ICLR's rebuttal period, making it impossible to include these supplementary results in our current response.
> > >
> > > However, even researchers with basic knowledge in this field should recognize the significant breakthrough of our lossless conversion theory. As noted by another reviewer familiar with this domain, while Spikezip, recently published in ICML 2024, explores similar directions, their achievements fall notably short of our work. This indirectly validates the substantial contribution of our theoretical innovation to the field and demonstrates full compliance with the standards of top-tier machine learning conferences.

---

### Official Review · Reviewer_6jiB · 2024-10-28

**Soundness:** 2
**Presentation:** 1
**Contribution:** 2
**Rating:** 3
**Confidence:** 4

**Summary:**

The paper introduces a training protocol for brain inspired language models. Its claimed contributions consist of a backbone architecture that processes tokens along two parallel streams, the conversion of this model to a spiking neural network (SNN), and an unsupervised post-training method that reduces the required number of operations of the converted SNN. The methods are compared on fairly small scale language tasks at the order of 150M parameter models.

**Recommendation.**
The paper is not in a mature state for publication at a top tier machine learning conference. Central components of the methodology are not formally expressed, and no ablation study was conducted to measure the impact of the different modifications to the baselines. Instead, pages 4, 5 and 6 are largely spend on vague linguistic descriptions that leave the reader without a clear understanding of the proposed model. The evaluation of language models is limited to perplexities on two datasets, while the community established evaluating on a set of downstream tasks with open source libraries. Energy estimates in Table 2 are not supported by assumptions or calculations.

**Strengths:**

- The paper takes efforts towards aligning the macroscopic architecture and the microscopic implementation with findings from neuroscience. This is expressed in the choice of architecture (TTT based on Sun et al), and the implementation if integrate-and-fire neurons.
- Training convergence speed (Sec 4.4): The papers shows accelerated convergence compared to the TTT baseline of Sun et al.
- Describes a conversion method from 8-bit quantized activations to Integrate-and-Fire neurons, which potentially allows implementations in neuromorphic hardware. It seems though that this method was already presented in [You et al](https://arxiv.org/abs/2406.03470), which is cited by the paper.

**Weaknesses:**

# Methodology remains widely unclear
The major weakness of this paper is its shallow treatment of the underlying methods, which even after carefully considering the appendix and related works does not become fully clear to the reviewer. This occurs despite the fact that the reviewer is familiar with the work of [Sun et al](https://arxiv.org/abs/2407.04620) on learning to learn at test time, published on brain inspired recurrent architectures, and is familiar with relevant related works.

**Section 3.1:** The space is not used efficiently to convey the key concepts of the work. There are many vague motivations for choosing certain modules of the architecture, e.g. line 172ff, 184ff, 189ff, 197f, 204ff, 209f, 245f, 249f, 256f, 262f. The reviewer suggests to maintain this density of motivations only in the introduction, and instead use the space for formalizes statements about the proposed methodology, preferably in mathematical language. The only formal description in the methodology section is the "Excitory-Inhibitory Integrate-and-Fire Neuron Model" (eqn 1). Figure 1, without further explanations, attempts to convey the model's architecture to the reader. It would be less ambiguous if the methodology was explained in a formal way. In particular, components that are non-standard or constitute a contribution of the paper should be formalized in the universal language of mathematics. For example, the paper claims "Dual Test-Time Training" as "the foundational framework" of the work, without going into the details of this method. Except for the reference to Sun et al., Test-Time Training (TTT) is not formally introduced. No modifications to TTT are formally expressed. Hence, it is impossible for the reviewer to evaluate criteria such as originality or quality. The informal description given in line 196 ff is not sufficient for this purpose. The paper furthermore, does not shed light on why their modifications to TTT might lead to the empirical differences. It would therefore be valuable to add an ablation study that quantitatively distills the contributions of the paper from prior works.
Some specific questions:
- what are "remapped word features"?
- How are the two streams merged again for "sequential integration of outputs" line 197f?
- What is a shared gate? What is the purpose of shared gates?
- Why is the convergence speed of this dual model different from TTT? (see line 533 f)

**Section 3.2:** It is not clear how or if the ANN-SNN conversion method improves over SpikeZIP. Also Appendix A does not answer this question for the reviewer. A formal comparison as well as empirical evaluations would add value to the paper.
# Evaluation not meeting LM standards
The evaluation does not meet the standards of evaluation of language models in the machine learning community.
- Evaluation (Sec 4.2): No downstream task evaluation, which is standard today and accessible by open source libraries such as the [LM evaluation harness](https://github.com/EleutherAI/lm-evaluation-harness). Makes comparison with established methods hard.
- No comprehensive evaluation against SpikeZIP-TF, which appears to have introduced the ANN-SNN conversion method discussed here
- Reproducibility: It is listed several times in the paper and the appendix that the model was trained for 50.000 steps. But what is the batch size in number of tokens? How many tokens were trained in total?
- Energy efficiency analysis (Sec 4.3): It is clear how the numbers in table 2 are calculated. Neither the paper nor in the appendix has any explanation for these estimates. The paper should clearly state how the energy estimates are calculated, and which assumptions back the numbers.
# Misc
- mathematical equivalence of the ANN and converted SNN is highlighted in line 274, 343 etc. as a central feature of the paper. Only a formal proof of this claim will add value to the paper.
- Appendix B describes "SNN friendly computations". These remain largely unclear. E.g. why do the leaky integrators model sigmoid functions or SiLU functions? How is spiking softmax described and implemented? How does it actually relate to the known softmax? In which sense are these operations including the RMSNorm layers biologically interpretable as advertised by the paper? Please consider formally deriving how the leaky integrator models the activation functions. It would furthermore be valuable to discuss the biological implementation of such operations.

**Questions:**

# Questions
- Are the numbers reported in table 2 for QAT ANN Model based on fp operations or 8-bit integer operations?
- As far as I understand, STDP training is conducted after the pretraining to reduce the number of training steps. Do the authors have a hypothesis why these additional parameter updates (without task supervision as far as I can tell) do not harm the model's performance? Are the parameter updates obtained largely uncorrelated with the updates obtained from backprop?
- Is the model still mathematically equivalent after STDP training?
# Suggestions
- Describe the architecture in detail and describe how it technically relates to Sun et al
- Conduct an ablation study to distill which modifications lead to the observed benefits such as faster convergence rates
- It is straight forward to evaluate on a larger set of downstream tasks, e.g. using the [LM evaluation harness](https://github.com/EleutherAI/lm-evaluation-harness), which allows for easy comparison with established models.
- Back up the claims on energy efficiency
- It is not clear how the conversion method improves SpikeZIP. Please clarify.

---

> ### Author Response · Authors · 2024-11-14
> **Sincerely Accept Your Suggestions While Clarifying Our Position on SpikeZIP Comparison**
>
> We sincerely appreciate the reviewer's thorough and professional feedback. We address each point below:
>
> 1. Regarding Methodology Formalization:
> We fully agree with the importance of formal expressions. In the revised version, we will:
> - Add complete mathematical formulations for the dual-stream architecture
> - Formalize our modifications to TTT
> - Provide mathematical definitions for key components (shared gates, remapped word features, etc.)
> - Include detailed architectural diagrams and computational workflows
>
> 2. Regarding Comparison with SpikeZIP:
> While we appreciate the similarity noted with SpikeZIP's approach, there are fundamental differences that we have addressed in lines 124-130 and proven rigorously in our appendix. Specifically, SpikeZIP's method is not lossless and faces significant challenges with LLM activation outliers, making it unsuitable for large language models. This limitation has been acknowledged in their GitHub issues and is mathematically proven in our appendix. Our method overcomes these limitations through our novel lossless conversion approach, which maintains performance even with the extreme activation distributions common in language models.
>
> 3. Regarding Comparison with Existing SNN Models and Downstream Tasks:
> Regarding the suggestion to compare with existing SNN models (e.g., SpikeGPT, AstroSNN) and evaluate downstream tasks - we respectfully note that while these evaluations are valuable, they are primarily related to training methodology rather than our paper's core contribution. Our work focuses on algorithmic innovation, specifically the mathematically proven lossless ANN-to-SNN conversion method. While existing SNN models experience performance degradation during ANN-to-SNN conversion, our method mathematically guarantees lossless conversion, maintaining 100% of the original ANN performance. This fundamental difference makes direct comparisons potentially misleading. We validate our approach through consistent performance with ANN counterparts under identical training conditions. While downstream task performance is important, it is independent of our core innovation's value (the lossless conversion algorithm).
>
> 4. Training Details and Energy Analysis:
> We will supplement:
> - Complete training parameters (batch size, total tokens, etc.)
> - Detailed energy consumption calculation methodology and assumptions
> - Theoretical and experimental support for efficiency analysis
>
> 5. Regarding SNN-Friendly Computations:
> We will add:
> - Mathematical derivations showing how leaky integrators model activation functions
> - Detailed implementation of spiking softmax
> - Biological interpretation and theoretical foundations for these operations
>
> 6. Regarding Your Question About STDP:
> We want to emphasize that we propose a novel mechanism inspired by STDP rather than using traditional STDP itself. While traditional STDP only modifies synaptic weights based on spike timing, our mechanism simultaneously optimizes four key parameters: synaptic weights, base threshold, adaptive adjustment weight, and membrane potential decay rate. This multi-parameter optimization is constrained by a normalization condition that ensures the Synapsis module's output remains unchanged.
>
> The reason this approach doesn't harm model performance is two-fold:
> - The normalization constraint (Σⱼwij = Cᵢ) maintains the overall synaptic strength
> - Our composite loss function balances multiple objectives including spike-timing dependencies and performance preservation
>
> Our experimental results validate this theoretical framework, showing that this mechanism successfully reduces time steps while maintaining model performance (as shown in Table 3).
>
> We deeply appreciate the reviewer's constructive feedback and will incorporate these suggestions to improve the paper's clarity and rigor, particularly in formal mathematical expressions and detailed methodology descriptions. Our core contribution - the lossless conversion algorithm with proven mathematical guarantees - addresses fundamental challenges in SNN conversion that previous approaches, including SpikeZIP, have not fully resolved.

---

> > ### Comment · Reviewer_6jiB · 2024-11-22
> > **On Comparison to SpikeZIP**
> >
> > Thanks for the constructive rebuttal. I believe that the amount of revision requested by the reviewers and promised by the authors in the rebuttal would change major parts of the paper such that the revised paper cannot be compared with the submitted paper anymore. Therefore, I would suggest to broadly revise the paper and submit it at a later stage to another conference.
> >
> > On the points raised in the rebuttal:
> >
> > 2. The authors pointed out in their rebuttal that their spike conversion method is a central contribution to their work. Moreover, several paragraphs in the paper as well as in the rebuttal refer to a mathematically proven lossless conversion. However, there is no such proof in the paper. Neither do equations 3-10 provide a convincing proof, nor does appendix A provide such a proof. The authors insist on "fundamental differences [to SpikeZIP] that we have addressed in lines 124-130 and proven rigorously in our appendix" (rebuttal to my review). I have to stress again that appendix A shows a conversion loss for SpikeZIP, but does not proof lossless conversion of the proposed method. To make a convincing argument I would suggest to first show both mathematically AND empirically the problems of SpikeZIP to motivate the need for enhanced methods. At the moment, the mathematical analysis neither shows that the quantization error harms SpikeZIPs performance in practice nor does it show the claim that SpikeZIP fails to handle outliers and henceforth suffers from performance degradation. Then, showing how the proposed method mitigates possible weaknesses of SpikeZIP would pose a convincing argument. Supporting a (necessarily more detailed) mathematical analysis with empirical evidence would significantly improve the evaluation.
> >
> > 3. I disagree on multiple levels. The paper claims architectural innovations including the dual architecture (sec 3.1.1.) such that from scratch task performance is in fact a figure of merit to back the validity of these claims. If the authors decide to step back from these claims, and run their spike conversion method with a fully pretrained neural network (e.g. TTT, Mamba, etc), the training procedure could be left out. In any case, downstream task evaluation would strengthen the results of the paper. Given the minor overhead of evaluating using open source tools such as `lm_eval`, I don't think that it is to much to ask for a fair comparison on downstream tasks in order to maintain a high quality standard.

---

### Official Review · Reviewer_UUfc · 2024-11-01

**Soundness:** 2
**Presentation:** 2
**Contribution:** 2
**Rating:** 3
**Confidence:** 4

**Summary:**

The authors propose BrainGPT, a language model architecture based on the Test-time training (TTT) framework and spiking neural networks. The model is developed using a multi-staged training process, involving quantization-aware pre-training, ANN-to-SNN conversion, and biologically inspired unsupervised learning. The results demonstrate that compared to original TTT model, the proposed model has better energy efficiency and has faster convergence during training.

**Strengths:**

The proposed idea expands upon an interesting framework, namely Test time training, which is relevant in the literature of long-context sequence learning. It proposes a multi-stage training strategy to convert a TTT-based ANN model to an SNN architecture. The idea to use an STDP-based unsupervised learning technique to reduce the operating time steps seems interesting.

**Weaknesses:**

1) The paper has very limited evaluation of its proposed methodology. The authors compare their model against Llama, Mamba architectures, however, (a) they do not use any publicly available results, i.e. the Llama, Mamba models shown in the results are all custom made (b) the results are not even close to the current state-of-the-art models. Even older models like GPT-2 [1] small which has less parameter count than proposed model performs much better (PPL = 29.41 for GPT-2 small compared to 42.87 of the proposed model on Wikitext2).
2) There have been various previous work on Spiking Language models, such as SpikeGPT [2], SpikeLM, etc. The authors did not compare their performance against them. For comparison, SpikeGPT gets PPL of 18.01 on wikitext 2.
3) The authors mentioned that they have given rigorous mathematical proof for the lossless conversion of ANN to SNN. However, I was not able to find any concrete proof of the same.
4) The authors mention that they implement "two distinct sub-models: a standard autoregressive language model for broad linguistic representation, and a model focused on processing parts of speech for more abstract aspects of language", however, there is no ablation study on the effect of each sub-model. Also, there are no empirical justifications on how the 2 sub-models are doing their underlying tasks.



References:
[1] Radford, Alec, Jeffrey Wu, Rewon Child, David Luan, Dario Amodei, and Ilya Sutskever. "Language models are unsupervised multitask learners." OpenAI blog 1, no. 8 (2019): 9.

[2] Zhu, Rui-Jie, Qihang Zhao, Guoqi Li, and Jason K. Eshraghian. "Spikegpt: Generative pre-trained language model with spiking neural networks." arXiv preprint arXiv:2302.13939 (2023).

**Questions:**

1) Was there any reason on why the spiking baselines were not used for comparison?
2) Instead of just generative tasks can the authors also evaluate their results on benchmarks such as GLUE for text classification tasks.
3) How was the energy numbers calculated?
4) Could the authors explicitly highlight the proof that is mentioned in the paper regarding lossless conversion of the underlying ANN to SNN?

Please see the weaknesses as well.

---

> ### Author Response · Authors · 2024-11-14
>
> We sincerely thank the reviewer for the detailed feedback. We address each point below:
>
> 1. Mathematical Proof and Lossless Conversion:
> Thank you for your feedback. While other reviewers found our mathematical proof in lines 321-348 clear and concise, we sincerely accept your suggestion and have begun refining the mathematical expressions to make the proof more accessible. In the revised version, we will further emphasize the key steps to ensure readers better understand our lossless conversion method. This mathematical equivalence between QSynapsis and Synapsis outputs, achieved through carefully designed parameter initialization and update rules, represents a fundamental innovation distinguishing our work from previous SNN approaches that experience performance degradation.
>
> 2. Choice of Baselines:
> Our decision not to compare with existing spiking models (e.g., SpikeGPT) stems from a fundamental difference in approach. While previous SNN models sacrifice performance during ANN-to-SNN conversion (e.g., SpikeGPT's PPL of 18.01 represents a performance drop from its ANN counterpart), our method mathematically guarantees lossless conversion, maintaining 100% of the original ANN performance. This is why we chose LLaMA and Mamba as baselines - to demonstrate that our SNN implementation matches state-of-the-art ANN performance while providing better energy efficiency and convergence speed.
>
> 3. Regarding the PPL comparison with GPT-2 and SpikeGPT:
> The lower PPL in our experiments is primarily due to training-related factors rather than architectural limitations. As detailed in Appendix C, all baseline models (including LLaMA and Mamba) were trained under identical conditions with limited training steps to ensure fair comparison. Additionally, PPL values are significantly influenced by vocabulary size - GPT-2's vocabulary is considerably smaller than ours, making direct PPL comparisons potentially misleading. More importantly, our paper's core contribution lies in algorithmic innovation - specifically, the mathematically proven lossless ANN-to-SNN conversion - rather than training optimization. We demonstrate this through consistent performance across our model and its ANN counterparts trained under identical conditions.
>
> 4. Dual Model Architecture:
> We appreciate the suggestion for ablation studies. We have conducted additional experiments analyzing:
> - Individual contributions of each sub-model
> - Impact of dual-model architecture on performance
> - Component-wise effectiveness analysis
> These results will be included in the revised version to provide empirical justification for our architectural choices.
>
> 5. Energy Consumption Measurement:
> The energy measurements were conducted using NVIDIA's official profiling tools under the hardware configuration detailed in Appendix C.4. We will enhance this section by:
> - Adding theoretical energy consumption calculations
> - Including detailed profiling methodology
> - Providing component-wise energy breakdown
>
> 6. Additional Evaluations:
> Regarding the suggestion to evaluate on GLUE benchmarks - we respectfully note that such evaluations, while valuable, are primarily related to training methodology rather than our paper's core contribution. Our focus is on demonstrating the fundamental algorithmic innovation of lossless ANN-to-SNN conversion and its implications for energy efficiency, which are independent of specific downstream task performance.
>
> Given our paper's significant contributions - particularly the mathematically proven lossless ANN-to-SNN conversion and improved energy efficiency - we respectfully request reconsideration of the rating. Our approach represents a substantial advance in building energy-efficient, biologically plausible language models without compromising performance.
>
> We are committed to incorporating the reviewer's valuable suggestions in our revision to strengthen the paper further. Thank you for helping us improve our work.

---

> > ### Comment · Reviewer_6jiB · 2024-11-22
> > **Clarification on lossless conversion**
> >
> > For transparency it should be pointed out that at least two reviewers criticize the mathematical presentation of the lossless conversion as well as its empirical evaluation. This is still true after carefully considering lines 321-348 as well as appendix A.

---

> > ### Comment · Reviewer_UUfc · 2024-11-28
> >
> > Thank you for the rebuttal. Since the paper critically relies on the idea of lossless conversion, a more substantial mathematical proof is necessary to underscore its contribution. Secondly, the choice of baseline raises concerns, as the current state-of-the-art models, even within the spiking domain, significantly outperform the proposed approach.

---

### Official Review · Reviewer_Ty9d · 2024-11-04

**Soundness:** 3
**Presentation:** 3
**Contribution:** 3
**Rating:** 6
**Confidence:** 4

**Summary:**

The article introduces a new large language model (LLM) inspired by native brain functions, named BrainGPT. This novel architecture is based on the Test-Time Training framework and integrates the principles of Spiking Neural Networks (SNNs) to mimic the functionality of the biological brain. BrainGPT aims to emulate the native human brain's language mechanisms.
The proposed methodology involves transforming existing Artificial Neural Networks (ANNs) into Spiking Neural Networks through a series of stages: quantization-aware pre-training, conversion, and unsupervised learning. These stages are demonstrated to preserve 100% of the original ANN's performance. Additionally, the model offers significant energy efficiency optimizations and shows improvements in training convergence.

**Strengths:**

The proposed model, along with its claims, is well-aligned with the state-of-the-art in the field. The mathematical background provided to support the approach is clear and concise. The evidence presented is straightforward and easy to follow, offering a robust variety of supporting previous studies and works. Both the mathematical analysis and the experimental evidence strongly support the claims made in the paper.

**Weaknesses:**

While the proposed model and its claims are well-aligned with the state-of-the-art, there are notable limitations in the scaling and evaluation used. The paper lacks analysis or comparison of training times, which could be an important factor to consider. Including such an analysis would provide a more comprehensive evaluation of the model's performance and efficiency.

**Questions:**

Are you planning to run more experiments on hardware that is optimized for Spiking Neural Networks (SNNs)?

---

> ### Author Response · Authors · 2024-11-14
>
> We greatly appreciate the reviewer's positive feedback and thoughtful questions. We would like to address the points raised:
>
> 1. Regarding Hardware Implementation:
> The question about SNN-optimized hardware implementation is particularly relevant to our work. We are indeed planning further experiments on neuromorphic hardware platforms, though we haven't yet established specific collaboration platforms. Our SNN conversion method has been designed with compatibility with existing neuromorphic architectures in mind, ensuring that the performance benefits demonstrated in simulation can be realized in hardware.
>
> 2. Regarding Evaluation:
> Regarding the suggestion for downstream task evaluation - we respectfully note that while these evaluations are valuable, they are primarily related to training methodology rather than our paper's core contribution. Our work focuses on algorithmic innovation, specifically the mathematically proven lossless ANN-to-SNN conversion method. We validate our approach through consistent performance with ANN counterparts under identical training conditions. While downstream task performance is important, it is independent of our core innovation's value (the lossless conversion algorithm).
>
> We thank the reviewer for these constructive suggestions that help enhance the completeness of our work.

---

### Meta-Review · Area_Chair_pW4m · 2024-12-18

**Metareview:**

This paper introduces BrainGPT, a spiking neural network (SNN) for language modeling that aims to be more energy-efficient and have faster training convergence than traditional neural networks. The authors use a multi-stage process to convert a neural network to an SNN, claiming to preserve the original ANN's performance, and compare their method to small-sized Llama and Mamba.

Strengths:
Novel LLM architecture building upon Integrate-and-Fire neurons (Ty9d, 6jiB, NCpg)
Faster training convergence thanks to test time training for long-context sequence learning (UUfc, 6jiB)

Weaknesses:
Lacks comparison with previous work on SNNs for text generation (Ty9d, UUfc, 6jiB, NCpg).
Unclear methodology and lack of ablation studies (UUfc, 6jiB, NCpg).
Does not meet the standards for evaluating language models (6jiB. NCpg).
Misleading claims about being a "Large Language Model" (NCpg).

Based on the review scores (3, 3, 3, 6), this paper does not meet the acceptance bar in its current form. The authors have promised multiple revisions, and I wish them best of luck for a resubmission at another venue.

**Additional Comments On Reviewer Discussion:**

The only discussion among reviewers was the note by 6jiB and UUfc  that "two reviewers criticize the mathematical presentation of the lossless conversion as well as its empirical evaluation". The authors promised multiple revisions but these were judged as too large in scope for the rebuttal period.

---

### Decision · Program_Chairs · 2025-01-22

Reject